# TrasMuon: Trust-Region Adaptive Scaling for Orthogonalized Momentum Optimizers

**Peng Cheng**[*‡]          **Jiucheng Zang**[†]          **Qingnan Li**[‡]          **Liheng Ma**[§]
**Yufei Cui**[‡]     **Yingxue Zhang**[‡]     **Boxing Chen**[‡]     **Ming Jian**[‡]     **Wen Tong**[‡]

## Abstract

Muon-style optimizers leverage Newton-Schulz (NS) iterations to orthogonalize updates, yielding update geometries that often outperform Adam-series methods. However, this orthogonalization discards magnitude information, rendering training sensitive to step-size hyperparameters and vulnerable to high-energy bursts. To mitigate this, we introduce TrasMuon (**T**rust **R**egion **A**daptive **S**caling **Muon**). TrasMuon preserves the near-isometric geometry of Muon while stabilizing magnitudes through (i) global RMS calibration and (ii) energy-based trust-region clipping. We demonstrate that while reintroducing adaptive scaling improves optimization efficiency, it typically exacerbates instability due to high-energy outliers. TrasMuon addresses this by defining a trust region based on relative energy ratios, confining updates to a stable zone. Empirical experiments on vision and language models demonstrate that TrasMuon converges faster than baselines. Furthermore, experiments without warmup stages confirm TrasMuon's superior stability and robustness.

## 1 Introduction

Optimizer choice remains a bottleneck for training modern foundation models, shaping convergence and stability at scale (DeepSeek-AI et al., 2025; OpenAI, 2026; Anthropic, 2026). In practice, heterogeneous and heavy-tailed/outlier updates can trigger loss spikes and narrow the stable learning-rate region (Behrouz et al., 2025; Kimi Team et al., 2025; Park et al., 2025). Diagonal adaptive methods (Adam/AdamW and variants) provide robust coordinate-wise magnitude control (Kingma & Ba, 2017; Loshchilov & Hutter, 2019; Pagliardini et al., 2025; Marfinetz, 2025), but do not exploit matrix-level update structure. Muon-style optimizers revisit matrix-structured updates via momentum orthogonalization (Bernstein, 2025; Jordan et al., 2024). However, orthogonalization mainly controls *geometry* and discards magnitude information, increasing sensitivity to step-size/warmup choices and vulnerability to bursty, axis-localized energy spikes (Behrouz et al., 2025; Kimi Team et al., 2025; Park et al., 2025).

We propose **TrasMuon** (**T**rust-**R**egion **A**daptive **S**caling for **Muon**), which preserves Muon-style structured mixing while stabilizing magnitudes via global RMS calibration and feature-wise relative-energy trust-region damping. For a matrix parameter $W \in \mathbb{R}^{d_{\text{out}} \times d_{\text{in}}}$, TrasMuon applies

$$\Delta W_t = -\hat{\eta}_t \, O_t^{\text{base}} \operatorname{diag}(c_t), \qquad c_t \in [c_{\min}, 1]^{d_{\text{in}}}. \tag{1}$$

Here $O_t^{\text{base}}$ is obtained by NS orthogonalization plus lightweight row-wise second-moment scaling (NorMuon-style) (Li et al., 2025). The RMS-calibrated step size $\hat{\eta}_t$ improves cross-layer comparability and reduces step-size sensitivity (Bernstein & Newhouse, 2025; Large et al., 2024). The clipping vector $c_t$ selectively suppresses high-energy feature axes while largely preserving the structured mixing factor, and we stabilize this signal via effective-time (schedule-free) weighting (Defazio et al., 2024).

**Contributions:** We introduce TrasMuon, which combines Muon-style mixing with feature-wise trust-region clipping. TrasMuon achieves faster early-stage convergence and improved stability,

---

[*]Equal contribution. Correspondence to `peng.cheng.hit@gmail.com`.

[†]Department of Combinatorics and Optimization, University of Waterloo, Waterloo, Canada.

[‡]Huawei Canadian Research Institute, Canada.

[§]McGill University & Mila-Quebec AI Institute, Canada.

exhibiting reduced sensitivity to the learning-rate magnitude and scheduling choices. In particular, it consistently mitigates loss spikes induced by heavy-tailed, axis-localized gradient bursts, while preserving the characteristic geometry of the Muon-style optimizers.

## 2 RELATED WORK

**Diagonal preconditioning and Adam-style optimizers.** Adam/AdamW and many refinements remain default baselines due to robust diagonal second-moment scaling (Kingma & Ba, 2017; Loshchilov & Hutter, 2019; Yuan et al., 2025; Pagliardini et al., 2025; Marfinetz, 2025; Shao et al., 2025; Gupta & Wojtowytsch, 2025). These methods stabilize training via coordinate-wise magnitude control but do not explicitly exploit matrix-level structure; TrasMuon instead keeps matrix-structured mixing and adds trust-region adaptive scaling.

**Beyond diagonal: block/matrix preconditioners and trust ratios.** Richer preconditioners (e.g., K-FAC, Shampoo, Adafactor) capture non-diagonal curvature structure with tractable approximations (Martens & Grosse, 2015; Gupta et al., 2018; Shazeer & Stern, 2018). Layerwise norm/trust-ratio scaling such as LARS/LAMB controls step magnitudes by comparing parameter and update norms (You et al., 2017; 2020). TrasMuon does not estimate curvature factors; it constructs a near-isometric mixing factor via orthogonalization and stabilizes magnitudes using RMS calibration plus a trust-region clipping.

**Orthogonalized directions and Muon-style updates.** Muon-style optimizers use Newton–Schulz iterations to approximate polar factors, yielding near-isometric directions that can improve Transformer training (Jordan et al., 2024; Bernstein, 2025). Related perspectives connect orthogonalized updates to modular/geometry-aware optimization and practical variants (Bernstein & Newhouse, 2025; Large et al., 2024; Pethick et al., 2025; Ahn et al., 2025; Kumar et al., 2025; Khaled et al., 2025; Riabinin et al., 2025; Li et al., 2025). TrasMuon builds on these directions but targets a distinct failure mode: bursty, axis-localized energy spikes, addressed via feature-wise relative-energy clipping and temporal smoothing.

## 3 METHODOLOGY

We propose **TrasMuon** (**T**rust-**R**egion **A**daptive **S**caling for **Muon**), which *factorizes* matrix updates into a structured mixing factor(illustrated in Appendix B) and feature-wise trust-region adaptive scaling(in Algorithm 1).

**Trust-region adaptive scaling.** We control bursty, axis-localized updates using the per-column relative-energy ratio $r_{t,j} = E_{t,j}/(E_t^{\mathrm{ref}} + \epsilon)$ and apply multiplicative damping, corresponding to an implicit trust-region constraint $r_{t,j} \lesssim \tau$ (tuned by $\alpha$ and optional trigger $k$). We measure column energy on pre-orthogonalization momentum $M_t$,

$$E_{t,j} = \sum_{i=1}^{d_{\mathrm{out}}} M_{t,ij}^2, \tag{2}$$

use a robust reference based on the median,

$$E_t^{\mathrm{cur}} = \mathrm{Quantile}_{0.5}(\{E_{t,j}\}), \tag{3}$$

$$E_t^{\mathrm{ref}} = \beta_E E_{t-1}^{\mathrm{ref}} + (1 - \beta_E)E_t^{\mathrm{cur}}, \tag{4}$$

which resists inflation by sparse bursts (Hampel et al., 1986; Huber, 1981). We then apply a smooth, damping-only clip

$$c_{t,j}^{\mathrm{raw}} = \frac{1}{1 + \alpha \log(1 + r_{t,j})}, \tag{5}$$

$$c_{t,j}^{\mathrm{clip}} = \mathrm{clip}\big(c_{t,j}^{\mathrm{raw}}, c_{\min}, 1\big), \tag{6}$$

(optionally only when $r_{t,j} > k$), and set $c_t$ by temporal smoothing (Appendix B).

**Schedule-free temporal smoothing.**    We smooth the instantaneous clip with EMA

$$c_t^{\text{ema}} = \beta_c c_{t-1}^{\text{ema}} + (1 - \beta_c)c_t^{\text{inst}}, \tag{7}$$

and optionally apply schedule-free averaging (Defazio et al., 2024) (default $\gamma_t = \eta$):

$$S_t = S_{t-1} + \gamma_t^2, \qquad C_t = C_{t-1} + \gamma_t^2 c_{t-1}^{\text{last}}, \tag{8}$$

$$c_t^{\text{avg}} = \frac{C_t}{S_t + \epsilon}, \qquad c_t = (1 - \rho)c_t^{\text{ema}} + \rho c_t^{\text{avg}}. \tag{9}$$

We cache $c_t^{\text{last}} \leftarrow c_t$ between gate updates to avoid bias when the raw clip is computed every $K$ steps. Convergence analysis can be seen in Appendix C.

## 4    EXPERIMENTS

TRASMUON is evaluated across four complementary settings: a short-run language-model training pilot under a fixed-budget protocol that examines early descent, extended-window behavior, and a minimal replication check, with full hyperparameters and additional plots reported in Appendix D; vision transformer training with a standard ViT recipe on ImageNet-100 under multi-seed evaluation, together with an additional column-burst stress test in Appendix E; physics-informed neural networks for Helmholtz under controlled random-ROI sampling shifts that induce reproducible nonstationarity, with step-size alignment and learning-rate sensitivity diagnostics provided in Appendix F; and a controlled diagnostic study with column-localized outlier injection, used to probe the energy-indexed feature-wise clipping mechanism and its boundary conditions, as documented in Appendix G.

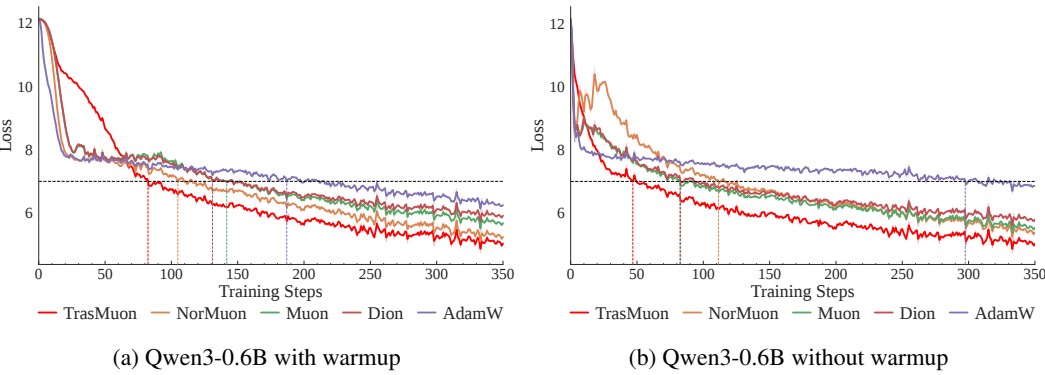

(a) Qwen3-0.6B with warmup                              (b) Qwen3-0.6B without warmup

Figure 1: Early-stage Qwen3-0.6B training dynamics from random initialization over steps 0–350, comparing warmup-enabled and warmup-free variants of the same warmup–stable–decay schedule. Curves are smoothed with a time-weighted EMA using smoothing factor 0.1 for visualization.

**Warmup-enabled dynamics.**    Under the warmup-enabled schedule, all evaluated optimizers train stably in this fixed-hyperparameter pilot. Although TRASMUON is not the fastest during the first few dozen steps, it begins to reduce loss more rapidly after approximately 80 steps, as shown in Fig. 1a. Using training loss 7.0 as a representative early-descent threshold,[1] TRASMUON reaches the target approximately $2.35\times$ faster than AdamW (80 vs. 188 steps) and $1.75\times$ faster than Muon (80 vs. 140 steps).

**Warmup-free calibration stress test.**    Without warmup, optimization becomes more sensitive to step-size calibration. Under the same shared learning rate and batch/sequence configuration, TRASMUON maintains a smoother early loss trajectory, while several baselines show larger oscillations, as shown in Fig. 1b. At the same reference threshold, TRASMUON reaches the target approximately $6.21\times$ faster than AdamW (48 vs. 298 steps) and $1.73\times$ faster than Muon (48 vs. 83 steps). Although warmup is widely used in language-model training, this setting remains informative because warmup length is typically chosen heuristically and can interact with optimizer-specific magnitude calibration.

---

[1]We use loss $= 7.0$ as a representative checkpoint for comparing early descent speed.

**Extended-window behavior under a fixed budget.** Beyond the early threshold-crossing window, TRASMUON also attains the lowest or comparable-lowest training loss in the extended window under both warmup settings in this pilot run. Extended-window curves and loss comparisons are reported in Fig. 2 of Appendix D.2. The gap between optimizers narrows as training enters a slower loss-reduction regime, suggesting that the observed advantage is most visible during the early, more nonstationary phase of training.

**Possible change in feature-wise energy concentration over training.** A plausible interpretation is that early training may exhibit stronger feature-wise anisotropy, where a small subset of hidden dimensions contributes disproportionately to gradient or momentum energy. In this regime, TRASMUON's relative-energy clipping can selectively damp axes with excessive update energy while preserving the structured Muon direction. As representations become better calibrated, energy may become more evenly distributed across feature axes, reducing the need for strong clipping and making the effective update closer to the NorMuon backbone. This explanation should be viewed as a hypothesis rather than direct evidence from the language-model runs; direct characterization of activation or gradient anisotropy, such as tracking energy ratios and gate statistics over time, is left to future work.

## 5    DISCUSSION AND LIMITATIONS

**What TRASMUON changes.** TRASMUON factorizes matrix updates into (i) a Muon-style near-isometric mixing factor constructed by Newton–Schulz orthogonalization and (ii) explicit magnitude controls: a global RMS-calibrated step size and a bounded, damping-only feature-wise clipping $c_{t,j} \in [c_{\min}, 1]$. This design targets a common practical tension: structured mixing can improve optimization geometry, while stable magnitudes govern learning-rate sensitivity and robustness to heavy-tailed bursts. When feature axes are semantically meaningful and bursts are axis-localized, TRASMUON selectively attenuates high-energy columns while largely preserving the Muon-style mixing structure.

**When feature-wise clipping and effective-time smoothing help.** Feature-wise clipping is most beneficial when update energy concentrates on a small subset of feature axes, reflected by large relative ratios $r_{t,j} = E_{t,j}/(E_t^{\mathrm{ref}} + \epsilon)$. In this regime, multiplicative clipping suppresses burst-dominated columns without amplification. When clipping is recomputed sparsely (every $K$ steps) or schedules vary, effective-time (schedule-free) averaging provides a stable long-horizon estimate by accumulating $\gamma_t^2$-weighted statistics, reducing sensitivity to recomputation frequency and schedule details.

**Limitations.** (i) The formulation is most natural for 2D weight matrices; extending energy diagnostics and damping to embeddings and higher-order tensors requires careful axis conventions. (ii) Newton–Schulz orthogonalization is sensitive to numerical precision; large-scale deployment benefits from precision-aware implementations. (iii) TRASMUON introduces additional hyperparameters and design choices (e.g., $K, \alpha, k, c_{\min}, \rho$ and the robust reference), and their interactions with model scale and data regimes merit broader sweeps.

## 6    CONCLUSION

We presented TRASMUON, a Muon-style optimizer that combines (i) NS-based near-isometric mixing factors with (ii) explicit magnitude stabilization via global RMS calibration and bounded, feature-wise trust-region clipping, optionally smoothed by effective-time (schedule-free) averaging. Across the evaluated workloads, TRASMUON improves training stability and achieves competitive or better final performance than strong baselines. Controlled diagnostics further support the intended mechanism: column-localized energy bursts increase relative energy ratios and lead to stronger applied damping, while ablations (e.g., NOCLIP) and broken-axis settings help rule out trivial explanations such as uniform step-size reduction. On practical tasks including language-model training, vision transformers, and PINNs under ROI-induced sampling shifts, TRASMUON yields faster or more stable optimization dynamics and improved robustness(Future Work can be deferred in Appencix A).

IMPACT STATEMENT

This paper presents work whose goal is to advance the field of machine learning. There are many potential societal consequences of our work, none of which we feel must be specifically highlighted here.

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

## A  FUTURE WORK

- **Generalizing feature axes beyond 2D matrices.** Extend EnergyCol-style clipping to convolutional kernels, embeddings, and higher-order tensors by defining principled "feature axes" (e.g., input-channel, attention-head, or group dimensions). We also plan to explore block-wise and low-rank variants that preserve interpretability while reducing per-step overhead.

- **Adaptive burst modeling with transparent control.** Replace fixed clipping hyperparameters (e.g., $c_{\min}$, $\alpha$, update period $K$) with lightweight, interpretable adaptations driven by online tail statistics of the energy distribution (quantiles, kurtosis, robust outlier scores), while maintaining the damping-only constraint and avoiding hidden amplification.

- **Broader robustness regimes and downstream impact.** Evaluate TRASMUON under realistic forms of nonstationarity common in large-scale training (curriculum shifts, domain-mixture changes, sequence-length spikes, and data-quality transitions), and study how clipping statistics correlate with downstream robustness and generalization.

## B  TRASMUON ALGORITHM (EXTENDED)

---

**Algorithm 1** TRASMUON: Muon + Adaptive Scaling + Trust Region + Schedule-Free Smoothed

---

**Input:** $W \in \mathbb{R}^{d_{\text{out}} \times d_{\text{in}}}$, base lr $\eta$, $\beta_1, \beta_2, \epsilon$, NS steps $T$, weight decay $\lambda$, $c_{\min}, \alpha, \beta_E, \beta_c$, trigger $k$, update period $K$, warmup $T_w$, mix $\rho$

Initialize $M \leftarrow 0$, $v^{\text{row}} \leftarrow 0$, $E^{\text{ref}} \leftarrow 0$, $c^{\text{ema}} \leftarrow \mathbf{1}$, $c^{\text{last}} \leftarrow \mathbf{1}$, $S \leftarrow 0$, $C \leftarrow \mathbf{0}$, $\gamma_t \leftarrow \eta$

**repeat**

  $G \leftarrow \nabla_W \mathcal{L}(W)$ ;   $W \leftarrow (1 - \eta\lambda)W$

  **Momentum:** $M \leftarrow \beta_1 M + (1 - \beta_1)G$

  **Orthogonalized direction:** $O \leftarrow \text{NS}(\tilde{M}; T)$

  $v^{\text{row}} \leftarrow \beta_2 v^{\text{row}} + (1 - \beta_2) \text{mean}_j(O_{\cdot j}^2)$

  $O^{\text{base}} \leftarrow \text{diag}\big((v^{\text{row}} + \epsilon)^{-1/2}\big) O$

  **Calibration:** $\hat{\eta} \leftarrow \eta \dfrac{\sqrt{d_{\text{out}} d_{\text{in}}}}{\|O^{\text{base}}\|_F + \epsilon}$

  **Column energy:** $E_j \leftarrow \sum_i M_{ij}^2$

  **Robust reference:** $E^{\text{cur}} \leftarrow \text{Quantile}_{0.5}(\{E_j\})$

  **EMA smooth:** $E^{\text{ref}} \leftarrow \beta_E E^{\text{ref}} + (1 - \beta_E)E^{\text{cur}}$

  **Schedule-free accumulators:** $S \leftarrow S + \gamma_t^2$, $C \leftarrow C + \gamma_t^2 c^{\text{last}}$

  $c^{\text{avg}} \leftarrow C/(S + \epsilon)$

  **if** $t > T_w$ **and** $t \bmod K = 0$ **then**

    $r_j \leftarrow E_j/(E^{\text{ref}} + \epsilon)$

    $c_j^{\text{clip}} \leftarrow \text{clip}\left(\frac{1}{1 + \alpha \log(1 + r_j)}, c_{\min}, 1\right)$

    **Trigger (optional):** $c_j^{\text{inst}} \leftarrow c_j^{\text{clip}}$ if $r_j > k$ else $1$

    **EMA smooth:** $c^{\text{ema}} \leftarrow \beta_c c^{\text{ema}} + (1 - \beta_c)c^{\text{inst}}$

  **end if**

  **Long-Short term Mixing:** $c \leftarrow \mathbf{1}$ if $t \leq T_w$ else $(1 - \rho)c^{\text{ema}} + \rho c^{\text{avg}}$

  $c^{\text{last}} \leftarrow c$ {cached between updates}

  **Update:** $W \leftarrow W - \hat{\eta}\left(O^{\text{base}} \odot \text{ExpandCols}(c)\right)$

**until** training ends

---

**TrasMuon** explicitly decouples *update geometry* (direction) from *step-size control* (magnitude). For a matrix parameter $W \in \mathbb{R}^{d_{\text{out}} \times d_{\text{in}}}$ with stochastic gradient $G_t = \nabla_W \mathcal{L}(W_t)$, it applies multiplicative coupling as

$$\Delta W_t = -\hat{\eta}_t O_t^{\text{base}} \text{diag}(c_t), \tag{10}$$

where $O_t^{\text{base}} \in \mathbb{R}^{d_{\text{out}} \times d_{\text{in}}}$ provides a structured direction, $\hat{\eta}_t$ is a row-wise RMS-calibrated global step size, and $c_t \in [c_{\min}, 1]^{d_{\text{in}}}$ is a *feature-axis* (column-wise) damping vector (damping-only, no amplification).

## B.1 Orthogonalized Directions via Newton–Schulz

TrasMuon maintains an exponential moving average of gradients

$$M_t = \beta_1 M_{t-1} + (1 - \beta_1) G_t. \tag{11}$$

To extract a rotation-robust, near-isometric direction, we approximate the polar factor of $M_t$. For numerical stability of Newton–Schulz (NS) iterations, we remove the scale gauge by RMS-normalizing

$$\tilde{M}_t = \frac{M_t}{\|M_t\|_F / \sqrt{d_{\text{out}} d_{\text{in}}} + \epsilon}, \tag{12}$$

and apply $T$ NS steps to obtain

$$O_t \approx \text{NS}(\tilde{M}_t; T) \approx \tilde{M}_t (\tilde{M}_t^\top \tilde{M}_t)^{-1/2}, \tag{13}$$

yielding a structured direction that is less sensitive to axis rotations than elementwise or diagonal preconditioning (e.g., Muon-style orthogonalized updates).

## B.2 Row-Second-Moment Scaling and RMS-Calibrated Step Size

Orthogonalization primarily shapes *direction*. To stabilize *magnitude* across layers and time, we apply lightweight row-wise second-moment scaling (as in NorMuon (Li et al., 2025)):

$$v_t^{\text{row}} = \beta_2 v_{t-1}^{\text{row}} + (1 - \beta_2) \operatorname{mean}_j \big( O_{t,\cdot j}^{\odot 2} \big), \tag{14}$$

$$O_t^{\text{base}} = \operatorname{diag}\big( (v_t^{\text{row}} + \epsilon)^{-1/2} \big) O_t. \tag{15}$$

Row scaling addresses row-wise heterogeneity, while a *global* row-wise calibration controls the update norm. We set

$$\hat{\eta}_t = \eta \cdot \frac{\sqrt{d_{\text{out}} d_{\text{in}}}}{\|O_t^{\text{base}}\|_F + \epsilon}, \tag{16}$$

so that the per-step RMS magnitude of $\Delta W_t$ is on the order of $\eta$. Since $c_t \leq \mathbf{1}$ elementwise, equation 16 also implies an explicit Frobenius-norm bound $\|\Delta W_t\|_F \leq \eta \sqrt{d_{\text{out}} d_{\text{in}}}$ (up to $\epsilon$), reducing sensitivity to layer shape and fluctuations in the orthogonalized direction (Bernstein & Newhouse, 2025; Large et al., 2024).

## B.3 Energy-Based Feature-wise Trust-Region Clipping

**Motivation.** In practice, instability often arises from *bursty magnitudes* that concentrate on a small subset of feature axes (columns), causing loss spikes and narrowing the stable learning-rate region. TrasMuon therefore introduces *feature-wise clipping*: it dampens only the high-energy feature directions while preserving the Muon-like direction structure in $O_t^{\text{base}}$.

**Column energy and a robust reference.** The column energy is measured from momentum $M_t$:

$$E_{t,j} = \sum_{i=1}^{d_{\text{out}}} M_{t,ij}^2, \qquad j = 1, \dots, d_{\text{in}}. \tag{17}$$

We summarize the typical energy level at step $t$ by a quantile statistic

$$E_t^{\text{cur}} = \text{Quantile}_q\big(\{E_{t,j}\}_{j=1}^{d_{\text{in}}}\big), \qquad q = 0.5, \tag{18}$$

and maintain a running reference via an EMA updated every step:

$$E_t^{\text{ref}} = \beta_E E_{t-1}^{\text{ref}} + (1 - \beta_E) E_t^{\text{cur}}. \tag{19}$$

Using a quantile (median) yields a high-breakdown reference: the sample median has a 50% breakdown point, so a sparse set of high-energy columns cannot arbitrarily inflate $E_t^{\text{ref}}$ and thereby "move the clipping threshold" in response to the outliers being clipped (Hampel et al., 1986; Huber, 1981).

**Relative ratio and clipping-style damping.** We define a dimensionless ratio as follow:

$$r_{t,j} = \frac{E_{t,j}}{E_t^{\text{ref}} + \epsilon}. \tag{20}$$

A hard energy cap $E_{t,j} \leq k\, E_t^{\text{ref}}$ gives the column-wise analogue of norm clipping:

$$c_{t,j}^{\text{hard}} = \min\left(1, \sqrt{\frac{k\, E_t^{\text{ref}}}{E_{t,j} + \epsilon}}\right), \tag{21}$$

which enforces $r_{t,j} \leq k$ after rescaling. TrasMuon leverages a smooth, numerically stable *soft clipping* rule:

$$c_{t,j}^{\text{raw}} = \frac{1}{1 + \alpha \log(1 + r_{t,j})}, \qquad c_{t,j}^{\text{gate}} = \text{clip}\left(c_{t,j}^{\text{raw}}, c_{\min}, 1\right). \tag{22}$$

This rule is bounded and avoids power-law instabilities as $r_{t,j} \to 0$. Importantly, $c_{t,j} \leq 1$ for all $j$, so the mechanism is strictly damping-only and can be interpreted as a trust-region safety mechanism in *feature space*.

**Triggered vs. continuous clipping.** Optionally, damping can be also applied only when $r_{t,j}$ exceeds a triggering threshold $k$:

$$\bar{c}_{t,j} = \begin{cases} c_{t,j}^{\text{gate}}, & r_{t,j} > k, \\ 1, & \text{otherwise,} \end{cases} \tag{23}$$

so that non-burst columns remain unchanged and the gate acts as an event-driven clip.

### B.4 TEMPORAL SMOOTHING AND SCHEDULE-FREE AVERAGING

To reduce short-term noise and avoid sensitivity to the gate-update period, we smooth the instantaneous clip by first applying EMA smoothing:

$$c_t^{\text{ema}} = \beta_c c_{t-1}^{\text{ema}} + (1 - \beta_c) c_t^{\text{inst}}, \tag{24}$$

where $c_t^{\text{inst}}$ denote the clip applied at step $t$ (either $c_t^{\text{gate}}$ or $\bar{c}_t$ depending on triggering). Second, we maintain a schedule-free average using an effective step weight $\gamma_t$ (default $\gamma_t = \eta$) (Defazio et al., 2024). Define the scalar accumulator $S_t \in \mathbb{R}$ and vector accumulator $C_t \in \mathbb{R}^{d_{\text{in}}}$:

$$S_t = S_{t-1} + \gamma_t^2, \qquad C_t = C_{t-1} + \gamma_t^2 c_{t-1}^{\text{last}}, \qquad c_t^{\text{avg}} = \frac{C_t}{S_t + \epsilon}, \tag{25}$$

where $c_{t-1}^{\text{last}}$ is the most recently applied clip (cached between updates). We then mix short- and long-term estimates:

$$c_t = (1 - \rho) c_t^{\text{ema}} + \rho c_t^{\text{avg}}, \qquad c_t^{\text{last}} \leftarrow c_t. \tag{26}$$

This effective-time averaging reduces sensitivity to warmup length and total training steps, and prevents bias when the raw clip is computed only every $K$ steps.

### B.5 FINAL TRASMUON UPDATE

Substituting the components into equation 10 yields the final update

$$\Delta W_t = -\hat{\eta}_t\, O_t^{\text{base}} \text{diag}(c_t), \tag{27}$$

which preserves Muon-like directional geometry via $O_t^{\text{base}}$ while controlling bursty magnitudes along feature axes through damping-only feature clipping $c_t$.

## C CONVERGENCE ANALYSIS

**Scope.** We provide a convergence *framework* for TrasMuon/energy clipping updates, separating (i) unconditional algebraic properties (bounded update norm; damping-only contraction) from (ii) mild alignment assumptions connecting the structured update to descent.

**Damping-only contraction.** For any matrix $A$ and any $c \in [0,1]^n$, right-multiplication by $\mathrm{diag}(c)$ cannot increase the Frobenius norm:

$$\|A\,\mathrm{diag}(c)\|_F \leq \|A\|_F. \tag{28}$$

**RMS calibration.** With $\hat{\eta}_t = \eta\sqrt{mn}/(\|O_t^{\mathrm{base}}\|_F + \epsilon)$ and damping-only $c_t \leq \mathbf{1}$, the update norm is uniformly bounded:

$$\|\Delta W_t\|_F \leq \eta\sqrt{mn} \qquad \forall t, \tag{29}$$

independently of transient gradient spikes.

**Stationarity under smoothness and alignment.** Under standard $L$-smoothness and a mild alignment condition (Appendix C), TrasMuon satisfies an expected first-order stationarity bound of the form

$$\frac{1}{T}\sum_{t=0}^{T-1}\mathbb{E}\|\nabla f(W_t)\|_F^2 \;\leq\; \frac{\mathbb{E}[f(W_0)] - f^\star}{\mu\,\eta\,T} \;+\; \frac{L}{2\mu}\,\eta\,mn, \tag{30}$$

for a constant $\mu > 0$ capturing effective descent. Importantly, the EMA/schedule-free construction of $c_t$ only affects how the clip is computed, while the theory relies solely on the invariant $c_{t,j} \in [c_{\min}, 1]$.

## C.1 ALGORITHMIC ABSTRACTION

Let $f : \mathbb{R}^{m \times n} \to \mathbb{R}$ and let $G_t$ be a stochastic gradient at $W_t$. We study updates of the form

$$W_{t+1} \;=\; (1 - \eta\lambda)\,W_t \;+\; \Delta W_t, \qquad \Delta W_t := -\hat{\eta}_t U_t, \qquad U_t := O_t^{\mathrm{base}}\mathrm{diag}(c_t), \tag{31}$$

with RMS-calibrated step size

$$\hat{\eta}_t \;=\; \eta \cdot \frac{\sqrt{mn}}{\|O_t^{\mathrm{base}}\|_F + \epsilon}, \tag{32}$$

and damping-only clipping

$$0 < c_{\min} \leq c_{t,j} \leq 1 \quad \forall j, t. \tag{33}$$

## C.2 DETERMINISTIC ALGEBRAIC PROPERTIES

**Lemma C.1** (Damping-only contraction). *For any $A \in \mathbb{R}^{m \times n}$ and any $c \in [0,1]^n$, $\|A\,\mathrm{diag}(c)\|_F \leq \|A\|_F$.*

*Proof.* $\|A\,\mathrm{diag}(c)\|_F^2 = \sum_{j=1}^n c_j^2 \|A_{\cdot j}\|_2^2 \leq \sum_{j=1}^n \|A_{\cdot j}\|_2^2 = \|A\|_F^2$. $\qquad\square$

**Lemma C.2** (Row-wise RMS calibration). *Under equation 31–equation 33, for all $t$,*

$$\|\Delta W_t\|_F \leq \eta\sqrt{mn}. \tag{34}$$

*Proof.* By Lemma C.1, $\|U_t\|_F = \|O_t^{\mathrm{base}}\mathrm{diag}(c_t)\|_F \leq \|O_t^{\mathrm{base}}\|_F$. Thus

$$\|\Delta W_t\|_F = \hat{\eta}_t\|U_t\|_F \leq \eta\sqrt{mn} \cdot \frac{\|O_t^{\mathrm{base}}\|_F}{\|O_t^{\mathrm{base}}\|_F + \epsilon} \leq \eta\sqrt{mn}.$$

$\qquad\square$

## C.3 ASSUMPTIONS FOR DESCENT

**Assumption C.3** ($L$-smoothness). $f$ is $L$-smooth with respect to Frobenius norm: $\|\nabla f(X) - \nabla f(Y)\|_F \leq L\|X - Y\|_F$.

**Assumption C.4** (Stochastic gradients). $\mathbb{E}[G_t \mid W_t] = \nabla f(W_t)$ and $\mathbb{E}[\|G_t - \nabla f(W_t)\|_F^2 \mid W_t] \leq \sigma^2$.

**Assumption C.5** (Alignment on the realized update). There exists $\mu_\Delta > 0$ such that for all $t$,

$$\mathbb{E}\big[\langle \nabla f(W_t), \Delta W_t\rangle \mid W_t\big] \;\leq\; -\mu_\Delta\,\eta\,\|\nabla f(W_t)\|_F^2, \tag{35}$$

where $\Delta W_t = -\hat{\eta}_t U_t$.

## C.4 EXPECTED STATIONARITY

**Lemma C.6** (Smoothness descent)**.** *Under Assumption C.3, for any $\Delta$,*

$$f(W_t + \Delta) \leq f(W_t) + \langle \nabla f(W_t), \Delta \rangle + \frac{L}{2}\|\Delta\|_F^2. \tag{36}$$

**Theorem C.7** (Expected stationarity for RMS-calibrated, damping-only updates)**.** *Assume C.3 and C.5 with $\lambda = 0$. Then for any $T \geq 1$,*

$$\frac{1}{T}\sum_{t=0}^{T-1} \mathbb{E}\|\nabla f(W_t)\|_F^2 \ \leq \ \frac{\mathbb{E}[f(W_0)] - f^\star}{\mu_\Delta\, \eta\, T} \ + \ \frac{L}{2\mu_\Delta}\, \eta\, mn, \tag{37}$$

*where $f^\star = \inf_W f(W)$.*

*Proof.* Apply Lemma C.6 with $\Delta = \Delta W_t$, take conditional expectation, use Assumption C.5, and bound $\|\Delta W_t\|_F^2$ by Lemma C.2. Sum over $t = 0, \ldots, T-1$ and telescope $f(W_t)$. $\qquad\square$

## C.5 EXTENSIONS: WEIGHT DECAY, STOCHASTICITY, AND PL

**Weight decay.** Decoupled weight decay can be handled by analyzing $f_\lambda(W) = f(W) + \frac{\lambda}{2}\|W\|_F^2$ or treating $(1 - \eta\lambda)W_t$ as an additional contraction term.

**Stochastic gradients.** Under Assumption C.4, the bound acquires an additional $\mathcal{O}(\eta\sigma^2)$ term as in standard SGD analyses.

**PL / strong convexity.** If $f$ satisfies the PL condition, one obtains linear-type convergence to an $\mathcal{O}(\eta mn)$ (or $\mathcal{O}(\eta\sigma^2)$) neighborhood.

# D SUPPLEMENTARY RESULTS: LANGUAGE-MODEL PRETRAINING

**Scope.** This appendix provides supplementary visualizations for language-model pretraining and a minimal replication check. The intent is *not* to introduce new claims beyond the main text, but to (i) document late-stage behavior under the fixed training budget and (ii) reduce the risk that qualitative trends are specific to a single random seed.

## D.1 EXPERIMENT SETTINGS

**Fixed-budget protocol and reporting (language model pretraining).** TRASMUON is evaluated in a controlled, short-run pretraining-style setting and compared against four baseline optimizers: AdamW (Loshchilov & Hutter, 2019), Muon (Jordan et al., 2024), Dion (Ahn et al., 2025), and NorMuon (Li et al., 2025). Decoder-only Transformer models are trained from random initialization, including GPT-2 (Radford et al., 2019) and Qwen3-0.6B (Yang et al., 2025), on FineWeb-Edu (Lozhkov et al., 2024). All runs follow the same fixed-budget protocol as Section 4: training proceeds for 1500 optimization steps with sequence length 1024 and a fixed global (effective) batch size of 1024, corresponding to $1500 \times 1024 \times 1024 \approx 1.57 \times 10^9$ training tokens. Unless otherwise stated, all optimizers share the same learning rate $\eta = 3.6 \times 10^{-3}$ and weight decay $\lambda = 5 \times 10^{-3}$, with no learning-rate sweep and no optimizer-specific retuning in this appendix. All runs follow the same default seeding behavior of the training stack. For NorMuon, the most influential optimizer-specific hyperparameter is the RMS normalization, which controls scale calibration of the orthogonalized update.

A warmup–stable–decay learning-rate schedule is adopted, and results are reported for two schedule variants: (i) a warmup-enabled configuration with 10% warmup and a final 20% decay phase, and (ii) the same schedule without warmup. All other training components, including the data pipeline, batching, tokenization, model architecture, and compute budget, are kept identical across optimizers. This section is treated as a fixed-hyperparameter pilot, where broader sweeps and multi-seed evaluations are deferred to future work.

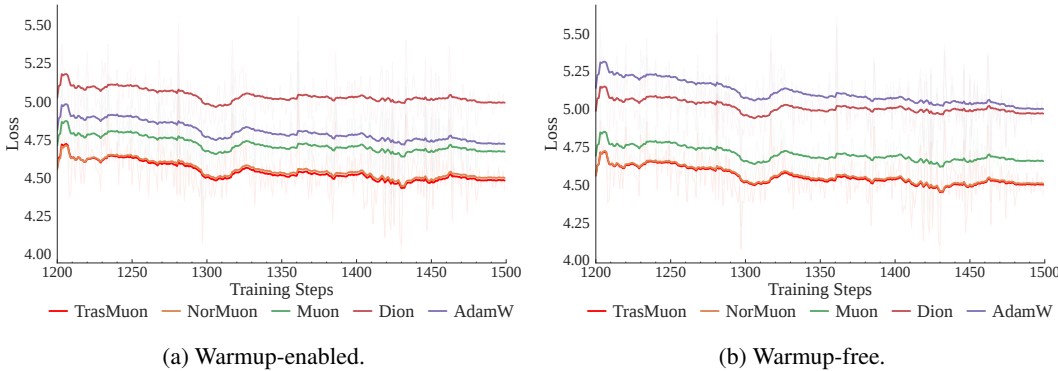

(a) Warmup-enabled.          (b) Warmup-free.

Figure 2: **Qwen3-0.6B late-stage window (steps 1200–1500) under a fixed-budget protocol.** Training loss under warmup-enabled vs. warmup-free variants of the same warmup–stable–decay schedule with a final decay phase near the end of training.

**Metrics.** Training loss is reported as a high-resolution indicator of early-stage optimization dynamics and stability under an identical training budget. This appendix presents the corresponding loss trajectories to complement the main-text summaries.

### D.2 QWEN3-0.6B: LATE-STAGE LOSS UNDER A FIXED 1500-STEP BUDGET

The late-stage training window (steps 1200–1500) is examined to complement the early-stage analysis in the main text. Figure 2 shows the corresponding loss trajectories in this window for both schedule variants. These curves document late-stage behavior under a fixed training budget and are not intended to imply full convergence.

### D.3 QWEN3-0.6B: MINIMAL REPLICATION CHECK

The main text reports Qwen3-0.6B results under a single fixed seed. To reduce the possibility that the observed qualitative trends are specific to that particular random draw, the same Qwen3-0.6B runs are repeated with an additional random seed while keeping *all* other settings identical, including the data pipeline, token budget, model architecture, learning rate, weight decay, and schedule variant. These replication runs are not used for hyperparameter tuning and are included solely as a robustness check under an identical protocol.

Figure 3 presents early-stage loss trajectories over steps 0–350 for the additional seed. Across both schedule variants, the qualitative optimizer behavior remains consistent with the main-text observations. No claim of statistical significance is made from this minimal replication; the results are provided as supplementary evidence of qualitative robustness under matched conditions.

### D.4 GPT-2 SMALL: ADDITIONAL ARCHITECTURE UNDER THE SAME PROTOCOL

GPT-2 Small is additionally evaluated under the same fixed-budget protocol to assess whether the observed qualitative dynamics transfer to a different architecture. Figure 4 presents early-stage loss trajectories over steps 0–350 for both schedule variants, and Figure 5 reports the late-stage window over steps 1200–1500. These plots are included to complement the main-text results, and the late-stage window is reported for completeness rather than as evidence of full convergence.

### D.5 LEARNING-RATE SWEEP ON QWEN3-0.6B

We further evaluate learning-rate sensitivity on Qwen3-0.6B in the same short-horizon language-model training pilot. While the main fixed-budget comparison follows the optimizer set used in the workshop experiments, this auxiliary sweep is designed to isolate learning-rate and schedule sensitivity. For this reason, we include Schedule-Free AdamW as an additional Adam-style baseline, since the sweep explicitly compares warmup-enabled and warmup-free schedules. Dion is omitted

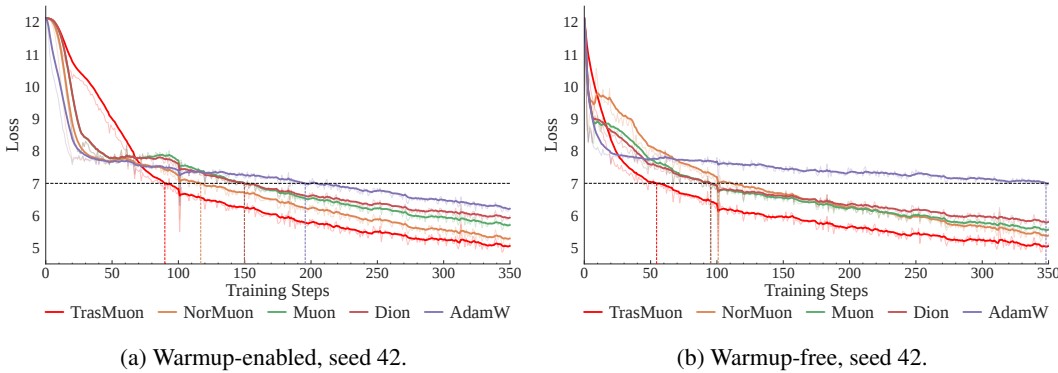

(a) Warmup-enabled, seed 42.

(b) Warmup-free, seed 42.

Figure 3: **Qwen3-0.6B replication under an identical protocol (additional seed).** Early-stage training loss (steps 0–350) for an additional random seed, under the same configuration as the main-text experiment.

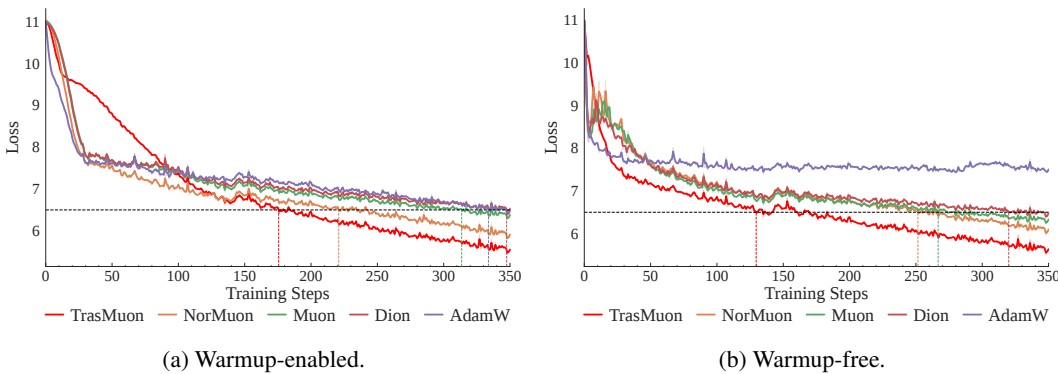

(a) Warmup-enabled.

(b) Warmup-free.

Figure 4: **GPT-2 Small early-stage dynamics (steps 0–350) under a fixed-budget protocol.** Warmup-enabled vs. warmup-free variants of the same warmup–stable–decay schedule.

from this diagnostic to keep the comparison focused on Adam-style magnitude adaptation, Muon-style orthogonalized updates, and schedule-free behavior under warmup removal. Thus, this sweep should be interpreted as a sensitivity analysis rather than a replacement for the main fixed-budget optimizer comparison.

The comparison includes AdamW, Schedule-Free AdamW, Muon, NorMuon, and TRASMUON. For each optimizer, we sweep four learning rates,

$$\{3.6 \times 10^{-4},\ 1.0 \times 10^{-3},\ 3.6 \times 10^{-3},\ 1.0 \times 10^{-2}\},$$

under both warmup-enabled and warmup-free schedules. Each entry in Table 1 reports the training loss at step 350. The purpose of this sweep is to check whether the short-horizon behavior is tied to a single shared learning rate, or whether it remains favorable after selecting the best learning rate separately for each optimizer and schedule setting.

Overall, TRASMUON achieves the lowest best loss in both schedule settings. With warmup, TRASMUON reaches the best loss of **5.009** at learning rate $3.6 \times 10^{-3}$, compared with 5.239 for NorMuon, 5.351 for Muon, 6.115 for AdamW, and 6.581 for Schedule-Free AdamW. Without warmup, TRASMUON again obtains the lowest best loss, reaching **5.017** at learning rate $3.6 \times 10^{-3}$, compared with 5.214 for NorMuon, 5.525 for Muon, 6.559 for AdamW, and 6.831 for Schedule-Free AdamW. These results suggest that the observed advantage of TRASMUON in this pilot setting is not only a consequence of using a single hand-picked learning rate.

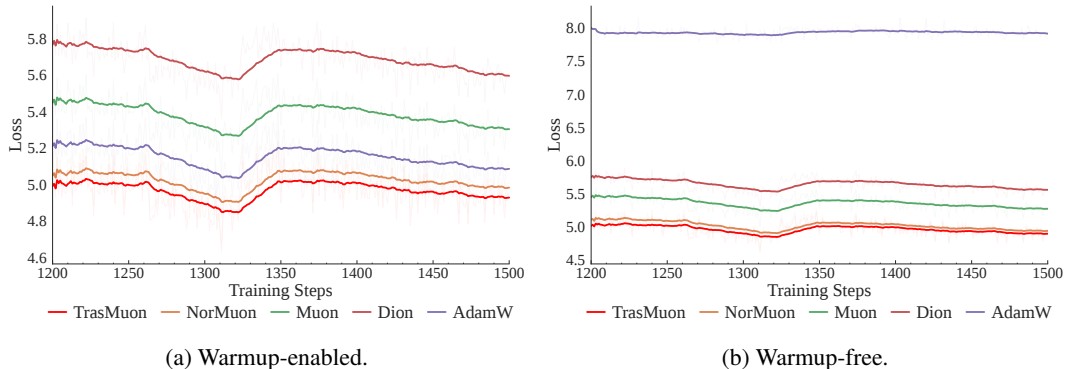

(a) Warmup-enabled.

(b) Warmup-free.

Figure 5: **GPT-2 Small late-stage window (steps 1200–1500) under a fixed-budget protocol.** Training loss for warmup-enabled vs. warmup-free schedule variants.

Table 1: **Learning-rate sweep for Qwen3-0.6B.** Training loss at step 350 is reported under warmup-enabled and warmup-free schedules. Lower is better. The best learning rate for each optimizer and schedule is shown in **bold**; the best TRASMUON entries are additionally underlined.

| | Warmup | | | | Warmup-free | | | |
|---|---|---|---|---|---|---|---|---|
| Optimizer | $3.6\times10^{-4}$ | $1.0\times10^{-3}$ | $3.6\times10^{-3}$ | $1.0\times10^{-2}$ | $3.6\times10^{-4}$ | $1.0\times10^{-3}$ | $3.6\times10^{-3}$ | $1.0\times10^{-2}$ |
| AdamW | 6.357 | **6.115** | 6.141 | 6.334 | **6.559** | 6.739 | 7.035 | 6.852 |
| Schedule-Free AdamW | 6.984 | 6.635 | **6.581** | 7.200 | 7.008 | **6.831** | 7.082 | 7.260 |
| Muon | 6.840 | 6.293 | 5.667 | **5.351** | 6.711 | 6.187 | **5.525** | 5.706 |
| NorMuon | 5.836 | 5.410 | **5.239** | 5.264 | 5.598 | **5.214** | 5.348 | 5.661 |
| **TRASMUON (Ours)** | 6.104 | 5.202 | **5.009** | 5.487 | 5.784 | 5.056 | **5.017** | 5.960 |

# E   VISION TRANSFORMER EXPERIMENTS

## E.1   IMAGENET-100 DATA SOURCE AND CONSTRUCTION

To reduce dataset preparation overhead, we use a publicly available ImageNet-100 *image archive* hosted on Kaggle (Ambityga, 2021) as a convenient storage source. Importantly, the Kaggle archive is used *only* as a source of image files. Class membership and the train/validation split are defined strictly according to the ILSVRC-2012 (ImageNet-1k) specification. Concretely, a fixed subset of 100 ILSVRC-2012 classes is selected (specified by synset IDs) and retained images whose labels match these synsets; we then follow the standard ILSVRC-2012 train/validation split. As a sanity check, we verify (in code) the synset-to-index mapping, per-class image counts, and that no images outside the selected synsets are included. This ensures that the benchmark corresponds to a well-defined subset of ImageNet-1k, independent of the hosting platform.

## E.2   IMAGENET-100 EXPERIMENTAL PROTOCOL AND SEED ALLOCATION

We evaluate optimization methods on ImageNet-100 using a ViT-Base/16 architecture at $224 \times 224$ resolution. Training follows a standard ViT/DeiT recipe (Dosovitskiy et al., 2021; Touvron et al., 2021), including random resized cropping, horizontal flipping, color jitter, RandAugment, and random erasing, together with label smoothing and Mixup (CutMix disabled).

Weight decay is applied with parameter grouping: LayerNorm and bias parameters use zero weight decay, and all remaining parameters use a fixed decay rate. Unless otherwise stated, all experiments use a base learning rate of $1 \times 10^{-3}$ and a weight decay of $5 \times 10^{-2}$. We compare AdamW (Loshchilov & Hutter, 2019), Muon (Jordan et al., 2024), NorMuon (Li et al., 2025), and TRASMUON under the same model architecture, data pipeline, training schedule, and compute budget, using three random seeds (42, 43, 44). Optimizer-specific parameters follow the respective published defaults and our unified implementation. For each method, we report the mean and standard deviation of *validation* top-1 accuracy across seeds (Table 2).

Table 2: **ViT on ImageNet-100.** Final *validation* top-1 accuracy (mean ± standard deviation) over three random seeds (42, 43, 44).

| Optimizer | Accuracy Mean | Accuracy Std |
|---|---|---|
| **TrasMuon** | **77.47%** | **0.34%** |
| NorMuon | 77.10% | 0.21% |
| Muon | 69.69% | 0.08% |
| AdamW | 42.53% | 4.38% |

### E.3 ImageNet-100: Vision Transformer Training.

We evaluate the benefits of TRASMUON in a large-scale vision setting by training ViT-Base (Dosovitskiy et al., 2021) on ImageNet-100 due to limited computational resources, a 100-class subset of ImageNet-1k (ILSVRC-2012) (Deng et al., 2009), followed the standard ILSVRC-2012 train/validation protocol. Dataset construction, class specification, and implementation details for ViT training are provided in Appendix E.1 and E.2. We compared AdamW, Muon, NorMuon, and TRASMUON, using identical training budgets and hyperparameters. Results show training loss and *validation* top-1 accuracy, aggregated over three random seeds.

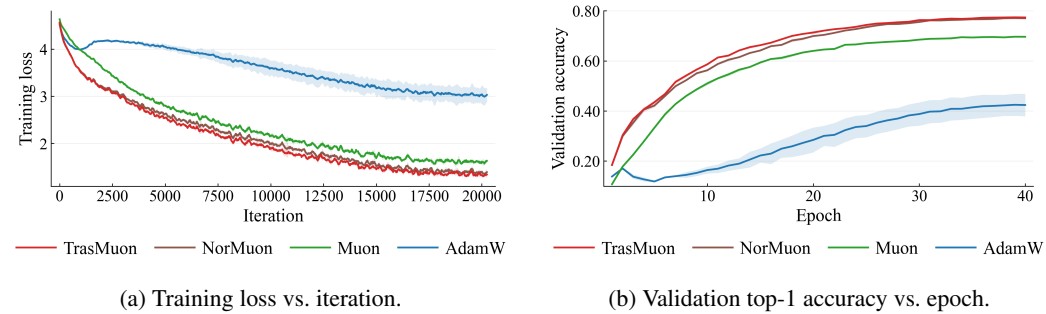

(a) Training loss vs. iteration.  (b) Validation top-1 accuracy vs. epoch.

Figure 6: ViT-Base training on ImageNet-100. Multi-seed results (mean ± std over three seeds: 42, 43, 44) for 4 optimizers. Shaded regions denote variability across seeds.

**Results.** Across all optimizers evaluated with multi-seed runs, Muon, Normuon, and TRASMUON consistently improve optimization behavior and validation accuracy over AdamW in these experiments, demonstrating the advantage of optimizers, which update based on structured, near-orthogonal update directions. illustrated in Fig. 6a and Fig. 6b, TRASMUON achieves the fastest loss reduction, the highest validation accuracy, and reduced variability across different seeds and various compared optimizers. Moreover, CIFAR-100 has been evaluated robustness under controlled column-localized burst injection in Appendix E.4.

### E.4 CIFAR-100: Column-Energy Stress Test

**Setup (stress-test benchmark).** We conduct a controlled stress test on CIFAR-100 using a Vision Transformer to assess optimizer robustness under axis-localized nonstationarity. Unless otherwise specified, all runs use 30 epochs, batch size 128, learning rate $1 \times 10^{-3}$, weight decay $5 \times 10^{-3}$, and identical data loading and preprocessing. We report mean and standard deviation of test accuracy over three random seeds (42, 43, 44).

**Burst injection.** To introduce structured nonstationarity without changing the data distribution, we inject sparse *column-localized gradient bursts* into selected large 2D parameter matrices (attention and MLP projections). The goal of this protocol is not to claim that injected bursts match the exact distribution of naturally occurring outliers, but to provide a reproducible mechanism-level stressor that concentrates energy along a small subset of feature axes. Crucially, the same burst pattern (targeted layers, selected columns, and timesteps) is applied across optimizers by fixing the burst

Table 3: **ViT on CIFAR-100 with column-localized gradient bursts.** Final top-1 test accuracy (mean $\pm$ standard deviation) over three seeds (42, 43, 44).

| Optimizer | Accuracy Mean | Accuracy Std |
|---|---|---|
| **TrasMuon** | **58.77%** | **0.22%** |
| NorMuon | 58.31% | 0.52% |
| Muon | 57.48% | 0.52% |
| AdamW | 35.03% | 5.05% |

random seed, enabling direct, fair comparison of optimizer responses. Full implementation details are provided in Appendix E.5.

**Observed behavior.** Table 3 summarizes test accuracy under burst injection. Across this stress setting, Muon-family optimizers maintain higher accuracy and lower variability than AdamW. Normalization-based variants reduce variance relative to Muon, while TRASMUON attains the highest mean accuracy and the smallest spread across seeds in this configuration.

### E.5 BURST INJECTION PROTOCOL IN CIFAR-100 BENCHMARK

We define a column-wise gradient burst operator applied to selected 2D weight matrices to induce controlled column-energy spikes without altering the data distribution. At each burst step, we select $k$ column indices (random or fixed, as specified) and perturb each selected column by adding a normalized random direction:

$$g_{:,j} \leftarrow g_{:,j} + \alpha \cdot \frac{u}{\|u\|_2 + \epsilon}, \quad u \sim \mathcal{N}(0, I).$$

The amplitude $\alpha$ can be specified as a fixed absolute value or scaled relative to the current gradient magnitude using a Frobenius-normalized reference:

$$\alpha = \rho \cdot \frac{\|g\|_F}{\sqrt{d_{\text{out}} d_{\text{in}}}},$$

optionally clipped by a maximum threshold. Bursts occur every $T$ optimization steps after an optional warmup phase and target only designated 2D layers (e.g., attention projections and MLP weights). Burst events and optimizer internal statistics (including feature-wise clipping coefficients, when applicable) are logged at the same timesteps to enable direct alignment between perturbations and optimizer responses.

## F PINNS WITH RANDOM-ROI SAMPLING STRESS TEST

Adaptive collocation in physics-informed neural networks (PINNs) is often necessary to resolve localized errors and stiff PDE behavior, where uniform sampling under-resolves difficult regions (Gao et al., 2023; Subramanian et al., 2022; Wu et al., 2023). Here we use region-of-interest (ROI) densification as a *controlled nonstationarity* mechanism to stress-test optimizer robustness: periodically concentrating interior collocation points in a small subregion induces distribution shifts in the residual samples, perturbing gradient statistics in a reproducible way.

**Helmholtz equation setup.** On $\Omega = [0, 1]^2$, we consider

$$\Delta u(x) + \kappa^2 u(x) = f(x), \qquad u|_{\partial\Omega} = 0, \tag{38}$$

with the manufactured solution

$$u^\star(x, y) = \sin(\pi k x) \sin(\pi k y), \qquad \kappa = \pi k. \tag{39}$$

which yields $f(x) = -(\pi k)^2 u^\star(x)$.

We train an MLP $u_\theta : \Omega \to \mathbb{R}$ by minimizing

$$\mathcal{L}(\theta) = \mathbb{E}\left[\tfrac{1}{2} r_\theta(x)^2\right] + \lambda_b \mathbb{E}\left[\tfrac{1}{2}\left(u_\theta(x) - u^\star(x)\right)^2\right] \tag{40}$$

Table 4: ROI patch pool used for ROI events (rectangles are $[x_0, x_1] \times [y_0, y_1]$).

| ROI patches |
| --- |
| Corners: $[0.00, 0.03] \times [0.00, 0.03]$, $[0.97, 1.00] \times [0.00, 0.03]$, $[0.00, 0.03] \times [0.97, 1.00]$, $[0.97, 1.00] \times [0.97, 1.00]$ |
| Edges: $[0.48, 0.53] \times [0.00, 0.05]$, $[0.48, 0.53] \times [0.95, 1.00]$, $[0.00, 0.05] \times [0.48, 0.53]$, $[0.95, 1.00] \times [0.48, 0.53]$ |
| Interior: $[0.20, 0.25] \times [0.20, 0.25]$, $[0.45, 0.50] \times [0.10, 0.15]$, $[0.10, 0.15] \times [0.55, 0.60]$, $[0.60, 0.65] \times [0.60, 0.65]$ |

$$r_\theta(x) = \Delta u_\theta(x) + \kappa^2 u_\theta(x) - f(x). \tag{41}$$

and report the relative error on a fixed evaluation grid

$$\text{rel-}L_2(u_\theta, u^\star) = \frac{\|u_\theta - u^\star\|_2}{\|u^\star\|_2}. \tag{42}$$

**ROI sampling protocol.**   To emulate adaptive densification, we impose a controlled, time-varying interior sampling distribution. We consider the Helmholtz equation on $\Omega = [0,1]^2$ with homogeneous Dirichlet boundary conditions and a manufactured solution $u^\star$. We run 4000 optimization steps and introduce nonstationary ROI events after step $t_0 = 1000$. At each step we sample $N_r = 1024$ interior points and $N_b = 256$ boundary points, with boundary weight $\lambda_b = 100$. We evaluate every 200 steps on a fixed $128 \times 128$ grid.

**Nonstationary ROI events (distribution shift).**   Starting from step $t_0 = 1000$, we trigger ROI events every $K_{\text{out}} = 20$ steps. At an ROI event step $t$, interior points are sampled from a mixture distribution

$$p_t(x) = (1 - \alpha) \, p_0(x) + \alpha \, p_{\text{roi}}^{(t)}(x), \qquad \alpha = 0.95, \tag{43}$$

where $p_0$ is uniform over $\Omega$ and $p_{\text{roi}}^{(t)}$ is uniform over the selected ROI patch $\Omega_{\text{roi}}^{(t)} = [x_0, x_1] \times [y_0, y_1]$. This yields repeated, time-varying distribution shifts that mimic practical ROI/adaptive refinement policies in PINNs.

**Random ROI patch pool (reproducible).**   To avoid sensitivity to a single ROI location, $\Omega_{\text{roi}}^{(t)}$ is chosen from a fixed pool (Table 4) using a deterministic `step_hash` seeding rule. Thus ROI locations vary across events while remaining fully reproducible given the experiment configuration and the training step index.

**ROI-local evaluation.**   To quantify localized disturbance and recovery, at ROI event steps we additionally compute an ROI-local rel$L_2$ on a $64 \times 64$ grid restricted to $\Omega_{\text{roi}}^{(t)}$. We estimate non-ROI error by sampling 4096 points from $\Omega \setminus \Omega_{\text{roi}}^{(t)}$. These diagnostics separate global convergence from localized behavior under distribution shifts.

### F.1    PINNs Benchmark: ROI Sampling as a Nonstationary Stress Test

**PINN ROI-sampling stress test: convergence and robustness.**   Figure 7a and 7b compare Muon and TRASMUON on Helmholtz ($k{=}2$) under a controlled nonstationary ROI-sampling protocol, where ROI events start at step 1000 and recur every 20 steps. During the initial stationary phase (before ROI events), both methods exhibit nearly identical optimization trajectories in terms of training loss and domain-wide relative $L_2$ error, indicating that TRASMUON does not incur a measurable overhead or degradation under standard uniform sampling.

After ROI events begin, the training objective becomes significantly more variable due to the induced distribution shifts in interior collocation points. In this nonstationary regime, TRASMUON maintains comparable or slightly lower training loss while exhibiting reduced extreme fluctuations, consistent with its design goal of suppressing bursty, feature-localized updates. These results support the conclusion that TRASMUON preserves baseline convergence under stationary sampling, while improving stability and final solution accuracy under controlled, nonstationary ROI sampling shifts.

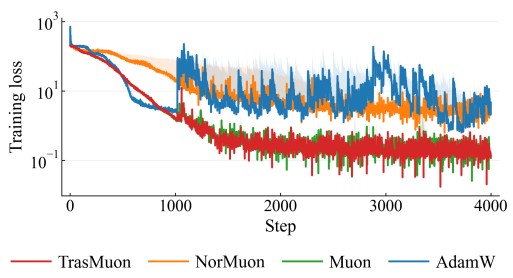
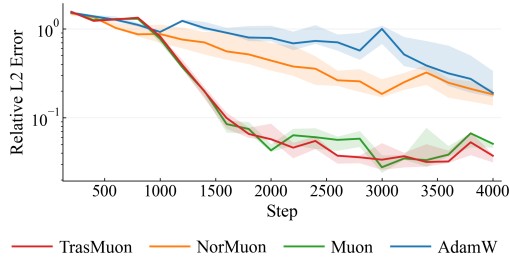

(a) training objective (estimated on the time-varying sampling distribution $p_t$)

(b) domain-wide relative $L_2$ error evaluated on a fixed grid. ROI events start at step 1000 and repeat every 20 steps

Figure 7: **PINN Helmholtz ($k$=2) under random ROI sampling shifts.** Curves show the mean over seeds, and shaded regions indicate variability across seeds.

### F.2 STEP-SIZE ALIGNMENT AND LEARNING-RATE SENSITIVITY (PINNs HELMHOLTZ, $k = 2$)

**Motivation.** Nominal learning rates are not directly comparable across update rules because different optimizers can induce different *effective* parameter-space step magnitudes. To reduce the confound that performance differences are driven by trivial step-size mismatches, we complement the main comparison with (i) a short step-size alignment diagnostic and (ii) a shared learning-rate (LR) sweep.

**Step-size alignment diagnostic.** Figure 8a reports the achieved effective step size, computed from parameter differences during an initial stationary window (before any ROI perturbations are introduced),

$$s_t = \frac{\|\Delta\theta_t\|_2}{\sqrt{P}}, \qquad P = \dim(\theta).$$

We target a fixed reference magnitude (dashed line) with a tolerance band (shaded region). Muon and TRASMUON attain comparable achieved step sizes within the tolerance range, supporting that subsequent robustness comparisons are not explained by a simple global step-size discrepancy.

**Learning-rate sweep.** Figure 8b shows the final relative $L_2$ error under a shared LR sweep. Shaded bands summarize variability across random seeds (median with an interquartile range). Both methods exhibit the expected degradation as LR increases beyond the stable region. Together with the alignment diagnostic, this sweep provides a complementary view of optimizer sensitivity to step-size calibration under the same training and sampling protocol.

### F.3 PINNs DIAGNOSTICS: METRIC DISTRIBUTIONS ACROSS SEEDS

We visualize the distribution of key robustness and accuracy metrics across random seeds for the PINN Helmholtz benchmark($k = 2$). This figure serves as a distributional check to ensure that the reported trends are not driven by a single favorable run.

## G CONTROLLED DIAGNOSTICS

This appendix provides protocol-level details and supporting evidence for the controlled diagnostics study. The intent is two-fold: (i) to make the stress protocol fully reproducible, and (ii) to document a minimal, time-aligned evidence chain that is *consistent with* the intended energy-indexed, feature-wise clipping mechanism under a controlled intervention. We do not introduce new claims beyond Section G.1.

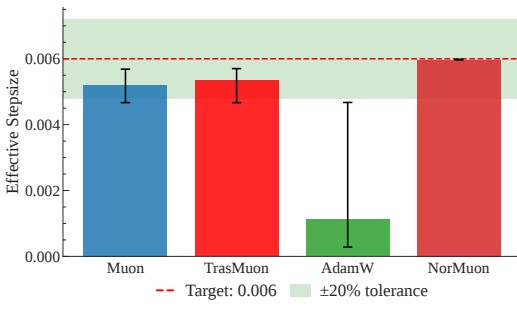
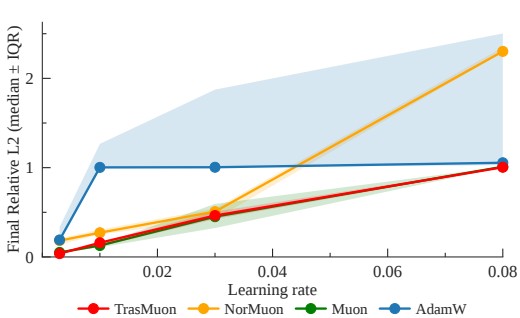

(a) **Achieved effective step size** during the stationary alignment window (before ROI events). The dashed line is the target, and the shaded region indicates tolerance. Error bars denote variability across seeds.

(b) **LR sensitivity** of final relative $L_2$ error under a shared LR sweep. Lines show the median across seeds and shaded bands indicate the IQR.

Figure 8: **PINNs Helmholtz ($k = 2$): step-size alignment and LR sensitivity.** (a) Effective step-size alignment reduces trivial magnitude confounds when comparing optimizers with different update rules. (b) A shared LR sweep summarizes sensitivity to step-size calibration via final relative $L_2$ error.

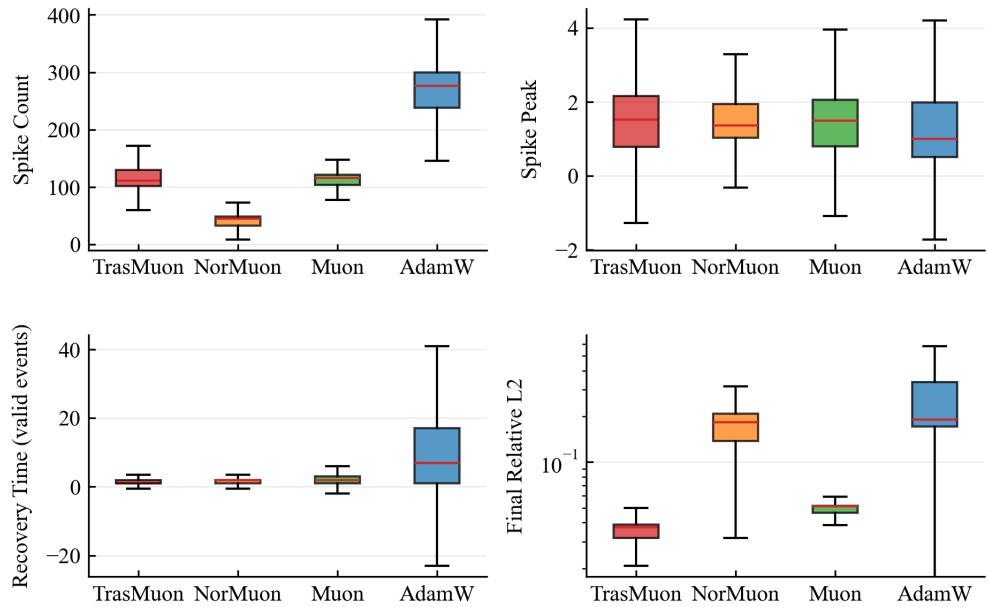

Figure 9: **PINNs Helmholtz($k = 2$): metric distributions across seeds.** Boxplots summarize spike count, spike peak, recovery time (valid events), valid-event rate, and final relative $L_2$ error across seeds under the same training protocol. Each box shows the median and interquartile range (IQR); whiskers indicate the remaining spread.

## G.1 CONTROLLED DIAGNOSTICS: COLUMN-LOCALIZED OUTLIERS AND ENERGY-BASED FEATURE CLIPPING

We design a controlled toy problem to validate the *feature-wise clipping* mechanism in TRASMUON under intermittent, column-localized bursts. We optimize a matrix parameter $W \in \mathbb{R}^{d \times d}$ under the quadratic objective

$$\min_W \; f(W) \; = \; \tfrac{1}{2}\|AWB - T\|_F^2, \tag{44}$$

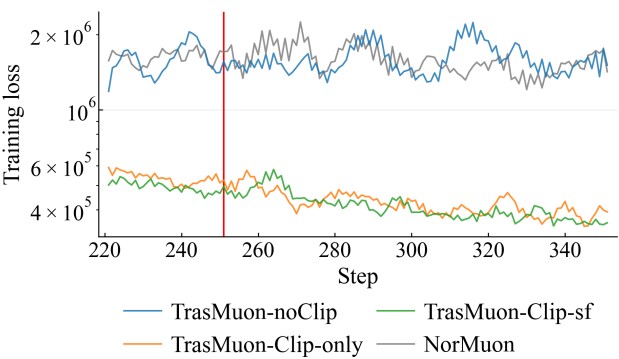

Figure 10: **Column outlier injection.** Loss trajectories in a window around an outlier event. Vertical markers indicate outlier steps.

where $A = U\Sigma_A U^\top$ and $B = V\Sigma_B V^\top$ with random orthogonal $U, V$, and diagonal spectra $\Sigma_A, \Sigma_B$ chosen to yield a target condition number $\kappa \in \{10^2, 10^4, 10^6\}$. This construction controls stiffness while allowing us to randomize nuisance rotations across runs.

**Column-localized outlier injection (stress protocol).** To emulate rare, feature-localized gradient domination, every $K_{\mathrm{out}}$ steps we inject an outlier event that amplifies a small subset of columns in a fixed feature basis. Concretely, for momentum $M_t$ we select a set $\mathcal{J}$ of $s \ll d$ column indices and apply a multiplicative burst

$$\widetilde{M}_{t,\cdot j} = \begin{cases} a\, M_{t,\cdot j}, & \text{if } j \in \mathcal{J}, \\ M_{t,\cdot j}, & \text{otherwise,} \end{cases} \tag{45}$$

with burst amplitude $a > 1$. This perturbation produces abrupt increases in column energy $E_{t,j} = \sum_i \widetilde{M}_{t,ij}^2$ while leaving the underlying objective equation 44 unchanged.

**Preserving feature semantics.** Because TRASMUON's clipping is axis-aligned (column-wise), the stress protocol is evaluated under a `fix_V=True` setting, i.e., the column basis is preserved across training and across injected events. We additionally report a boundary condition where the column basis is randomized (`fix_V=False`); in that case, injected energy disperses across columns and feature-wise clipping is not expected to yield an advantage.

**Metrics.** We track (i) spike count and (ii) final objective value, reporting median and IQR over multiple seeds/rotations.

**Closed-loop response.** Figure 10 shows that TRASMUON reduces burst-induced loss spikes and improves convergence relative to the NorMuon backbone under matched compute. Figure 11 provides mechanism-level evidence consistent with a closed-loop response: outlier events increase the relative column-energy ratio (e.g., $r_{q95}/r_{\max}$), which is immediately followed by a decrease in the *applied* clipping signal (tracked by $c_{\mathrm{used,min}}$), thereby damping the burst and suppressing spikes.

**Not a trivial step-size reduction.** To rule out the confound that improvements arise from a global effective step-size change, we include a TRASMUON-NOCLIP ablation that disables feature-wise clipping while keeping all other components identical. As summarized in Table 5, TRASMUON-NOCLIP behaves similarly to NorMuon, whereas enabling clipping yields a clear reduction in spike statistics and a large improvement in the final objective, isolating the contribution of feature-wise clipping.

**Boundary condition (feature semantics broken).** When the column basis is randomized, the advantage of feature-wise clipping diminishes, consistent with the intended mechanism: axis-aligned clipping requires a meaningful feature basis.

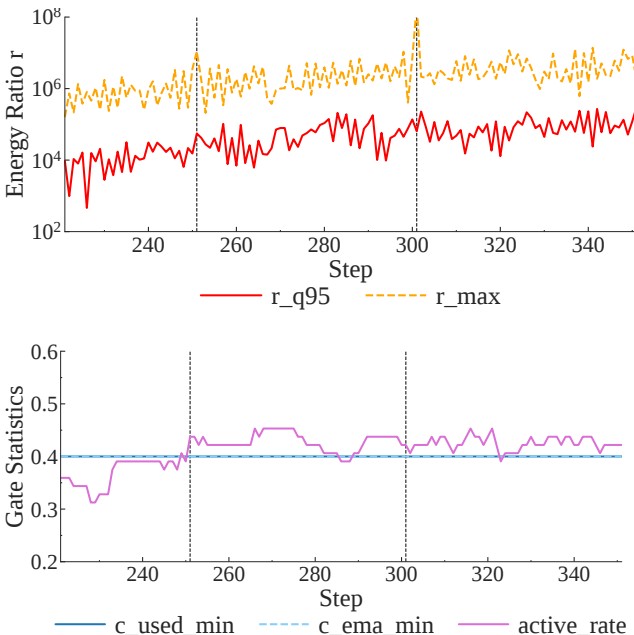

Figure 11: **Closed-loop clipping evidence.** Outlier events increase the column-energy ratio in log-scale (top), followed by stronger *feature-wise clipping* in the applied coefficients (bottom; $c_{\text{used,min}}$).

Table 5: **Controlled diagnoise summary under `fix_V=True`.** TRASMUON-NOCLIP removes feature-wise clipping while keeping the rest identical. We report median with IQR across runs; lower is better.

| Method | Spike Count | Final Loss |
|---|---|---|
| NorMuon | 44 (35,56) | 1.3e+06 (1.0e+06,1.6e+06) |
| TRASMUON-noClip | 48 (38,56) | 1.1e+06 (8.9e+05,1.9e+06) |
| TRASMUON-Clip-only | **28** (24,34) | 2.4e+05 (2.0e+05,2.8e+05) |
| TRASMUON-Clip-sf | 30 (24,36) | **2.0e+05** (1.6e+05,2.7e+05) |

### G.2 MINIMAL EVIDENCE CHAIN (TIME-ALIGNED OBSERVABLES)

Section G.1 argues that spike suppression is consistent with the following ordering under a controlled intervention: *outlier injection → relative energy increases → stronger applied clipping → reduced loss spikes*. We summarize the corresponding observables, which are directly logged and visualized in Fig. 11.

**(1) Outlier injection increases relative energy.** Eq. equation 45 increases the column energies $E_{t,j} = \sum_i \widetilde{M}_{t,ij}^2$ for $j \in \mathcal{J}$. We track the relative energy ratio

$$r_{t,j} = \frac{E_{t,j}}{E_t^{\text{ref}} + \epsilon}, \tag{46}$$

where $E_t^{\text{ref}}$ is the running reference used by TRASMUON. We visualize robust summaries such as $r_{q95}$ and $r_{\max}$, which rise at injected outlier steps (Fig. 11, top).

**(2) Higher relative energy is followed by stronger applied clipping.** TRASMUON produces damping-only clipping coefficients $c_{t,j}^{\text{used}} \in [c_{\min}, 1]$ that decrease with $r_{t,j}$. Consistent with this design, outlier steps are followed by a decrease in the applied signal, visible via $c_{\text{used,min}} = \min_j c_{t,j}^{\text{used}}$ (Fig. 11, bottom). This time alignment is consistent with the intended ordering "energy rise → clipping increase" under the intervention.

**(3) Applied clipping attenuates column updates (selectively).** Given the matrix-form update,

$$\Delta W_t \;=\; -\,\hat{\eta}_t \, O_t^{\text{base}} \, \text{diag}(c_t^{\text{used}}), \tag{47}$$

each column update magnitude is scaled by $c_{t,j}^{\text{used}}$:

$$\|\Delta W_{t,\cdot j}\|_2 \;=\; c_{t,j}^{\text{used}} \, \hat{\eta}_t \, \|O_{t,\cdot j}^{\text{base}}\|_2 \;\leq\; \hat{\eta}_t \, \|O_{t,\cdot j}^{\text{base}}\|_2. \tag{48}$$

Thus clipping is *selective*: only columns with $c_{t,j}^{\text{used}} < 1$ are damped, while the structured direction $O_t^{\text{base}}$ is preserved.

**(4) Spike suppression and objective improvement.** Consistent with (1)–(3), TRASMUON reduces loss spikes around outlier events (Fig. 10) and achieves lower final objective values than the backbone under matched compute (Table 5). Spike metrics (count/peak) are computed using the same deterministic detection rule across methods; details are provided in our experiment scripts and plotting utilities.

Table 6: **Boundary condition effects(`fix_V=False`).** When feature/column semantics are broken by column-space mixing, the advantage of feature-wise clipping diminishes.

| Method | Spike Count | Final Loss |
|---|---|---|
| NorMuon | 79 (74,86) | **1.1e+06** (8.4e+05,1.6e+06) |
| TRASMUON-noClip | 80 (74,87) | 1.4e+06 (9.8e+05,1.7e+06) |
| TRASMUON-clip-only | 74 (67,80) | 1.5e+06 (1.2e+06,1.8e+06) |
| TRASMUON-clip+SF | **72** (65,80) | 1.3e+06 (9.4e+05,1.7e+06) |

**Controls that break the chain.** We include two controls that remove key requirements of the mechanism: (i) TRASMUON-NOCLIP sets $c_t^{\text{used}} \equiv \mathbf{1}$, removing the attenuation in Eq. equation 48; empirically it behaves similarly to NorMuon (Table 5). (ii) Under `fix_V=False`, the injected energy is dispersed across columns, so clipping is no longer aligned with injected directions; correspondingly, the advantage of feature-wise clipping diminishes (Table 6).

