# OpenReview forum: "TrasMuon: Trust-Region Adaptive Scaling for Orthogonalized Momentum Optimizers"
_ICLR.cc/2026/Workshop/Sci4DL — Sci4DL 2026_

### Official Review · Reviewer_fpX8 · 2026-02-08

**Fit:** 3
**Significance:** 2
**Confidence:** 3

**Summary:**

This paper proposes TrasMuon, a trust-region adaptive scaling variant of Muon-style optimizers. The method augments Newton–Schulz–based orthogonalized momentum updates with global RMS calibration and feature-wise (column-wise) energy-based clipping, aiming to stabilize update magnitudes while preserving the near-isometric geometry of Muon. The key idea is to define a robust, relative-energy trust region that selectively damps bursty, axis-localized updates, combined with temporal smoothing and optional schedule-free averaging. Experiments on language models, vision transformers, and physics-informed neural networks demonstrate faster early-stage convergence and improved stability compared to Muon and Adam-style baselines, especially in warmup-free or highly nonstationary settings.

**Strengths:**

(1) Well-motivated problem formulation: The paper identifies a practical and underexplored limitation of Muon-style orthogonalized optimizers, namely the loss of magnitude information and vulnerability to axis-localized energy bursts, which is highly relevant to large-scale and nonstationary training.

(2) Principled and interpretable method design: TrasMuon cleanly separates update direction (orthogonalized momentum mixing) from update magnitude (adaptive scaling with a trust-region mechanism), resulting in a method that is conceptually clear and easy to reason about.

(3) Robust trust-region mechanism: The proposed feature-wise, relative-energy-based damping uses median normalization and a damping-only constraint, effectively suppressing unstable localized updates while preserving global geometric structure.

(4) Solid empirical validation: Experiments span multiple domains (language modeling, vision transformers, and PINNs) and include stress tests targeting axis-wise instability, demonstrating consistent improvements in stability and early-stage convergence over strong baselines.

(5) Clear presentation and reproducibility: The paper is well organized, with clear algorithm descriptions, informative ablations, and sufficient implementation details that support reproducibility and understanding.

**Suggestions:**

(1) From the update formulation, the main update takes the form $ O diag(c) $, i.e., only column-wise scaling of $ O $ is applied. A natural question is why one does not consider a more general form such as $ diag(c_1)O diag(c_2) $, where both left and right scalings are allowed. I believe this direction could be worth further exploration.

(2) I fully agree with the authors’ observation that using only $ O $ discards magnitude information. In this context, it would be informative to include a comparison with adaMuon, which performs element-wise scaling on $ O $.

(3) Personally, I am not fully convinced by Equations (2)–(9), as they appear somewhat overly complex and not particularly elegant in presentation.

---

### Official Review · Reviewer_819W · 2026-02-26

**Fit:** 1
**Significance:** 2
**Confidence:** 2

**Summary:**

The authors propose TrasMuon, an improved version of the Muon optimizer. Magnitude stabilization via global RMS calibration, feature-wise trust-region clipping, and schedule-free averaging of the clipping are added to default Muon. The authors provide experimental results for training from scratch on Qwen3-0.6B. Additional theoretical and empirical results are presented in the appendix.

**Strengths:**

The contribution and motivation are clear. The empirical results are convincing.

**Suggestions:**

How do the individual components of TrasMuon contribute to the empirical improvements? An ablation of applying scaling and clipping (with and without smoothing) would be valuable.

How were the hyperparameters chosen? You are introducing many new hyperparameters, and it is not clear if the empirical improvements can be observed across different training scenarios using stable hyperparameters.

I think the paper doesn't really fit the workshop. I acknowledge the contribution in engineering an improved optimizer; however, I don't see much contribution in revealing formerly unknown phenomena or explaining existing ones. The paper mainly combines known techniques to heuristically improve performance.

---

### Meta-Review · Area_Chair_quzi · 2026-03-01

**Recommendation:** Accept

**Metareview:**

The paper's claims are supported through empirical analysis and the proposed optimizer leads to improved and more stable optimization dynamics. I recommend acceptance.

---

### Decision · Program_Chairs · 2026-03-02

Accept